# Self-Supervised Steering and Path Labeling for Autonomous Driving

**DOI:** 10.3390/s23208473

**Published:** 2023-10-15

**Authors:** Andrei Mihalea, Robert-Florian Samoilescu, Adina Magda Florea

**Affiliations:** Department of Computer Science, Faculty of Automatic Control and Computers, University Politehnica of Bucharest, 060042 Bucharest, Romania; robert.samoilescu@stud.acs.upb.ro (R.-F.S.); adina.florea@upb.ro (A.M.F.)

**Keywords:** self-supervised learning, autonomous driving, steering prediction, semantic segmentation, steering geometry

## Abstract

Autonomous driving is a complex task that requires high-level hierarchical reasoning. Various solutions based on hand-crafted rules, multi-modal systems, or end-to-end learning have been proposed over time but are not quite ready to deliver the accuracy and safety necessary for real-world urban autonomous driving. Those methods require expensive hardware for data collection or environmental perception and are sensitive to distribution shifts, making large-scale adoption impractical. We present an approach that solely uses monocular camera inputs to generate valuable data without any supervision. Our main contributions involve a mechanism that can provide steering data annotations starting from unlabeled data alongside a different pipeline that generates path labels in a completely self-supervised manner. Thus, our method represents a natural step towards leveraging the large amounts of available online data ensuring the complexity and the diversity required to learn a robust autonomous driving policy.

## 1. Introduction

Driving a vehicle is a difficult task that requires complex reasoning and visual perception. The high level of stochasticity in the environment structure and dynamics leads to an intractable state space for traditional robotics trajectory planning approaches; thus, developing a system that can drive safely and can generalize to a wide variety of environmental setups remains an open problem. In 1989, the proposal of [1] to build the Autonomous Land Vehicle in which the traditional trajectory planning module is replaced by an entirely learnable process (end-to-end learning), showed that a neural network could exhibit high-level hierarchical reasoning—enough to steer a vehicle. Unfortunately, two major limitations of current end-to-end solutions consist of the lack of diversity and the reduced amount of available training data. In a recent study, the authors of [2] analyzed the impact of data (hours of driving) on the performance of a driving policy (distance to intervention), indicating a positive correlation. The results suggest that increasing the data diversity is likely to increase the model’s performance, thus bringing the promise that with enough data and computational resources, end-to-end solutions will yield good results.

In addition to extremely time-consuming human annotation processes for tasks such as semantic segmentation [3] or object detection [4], an autonomous driving dataset acquisition system requires expensive hardware. The need for additional sensors represents a bottleneck for collecting a geographically diverse and complex dataset. Recent developments in deep unsupervised learning have managed to avoid the need for LIDAR sensors, recovering depth information [5] and optical flow [6] from a single RGB camera with high accuracy.

Prior work has been focused primarily on the supervised paradigm to learn a driving policy [7] or path [8], from the sensory acquired signals.

We present two self-supervised methods to label an autonomous driving dataset for steering control and path segmentation. We combined existing work on path labeling, previously conducted in a weakly-supervised manner, using data from the camera, IMU, and LIDAR [9], with the recent advancements made in the field of predicting ego-motion from videos [10] and obtained a completely self-supervised pipeline that can be used to extract the steering and the path followed by a vehicle for each frame in a video. Our labeling method requires only a pose estimation network jointly trained with a monocular depth estimation one in a fully self-supervised manner. Even though driving datasets with ground-truth labels for the steering and ego-motion values are available, they still represent a very small fraction when compared to all of the unlabeled videos that can be utilized if a reliable method for extracting the labeled data would exist. Only using existing labeled datasets may limit the abilities of the methods trained on this data, when it comes to generalization on a broader spectrum of scenarios. Using our proposed framework, driving videos in the wild can be enriched with valuable, reliable data that can be used for different learning problems. We obtained competitive results against the models trained on the ground-truth sensory acquired labels for both tasks, proving the feasibility and robustness of our methods.

The main contributions of our work are:Propose a heuristic method for estimating the scaling factor of the vehicle ego-motion;Generate steering labels starting from predicted ego-motion estimations made by a network trained in a self-supervised manner;Generate trajectory segmentation labels from the same ego-motion estimations;Train a steering neural network on the generated steering labels and ground truth steering labels and compare the results;Provide a method for improving the steering commands by using the predicted trajectory labels.

## 2. Related Work

Recent research for autonomous driving has been focused primarily on the end-to-end paradigm, after the proposal of and introduction of ALVINN [1], one of the first vehicles driven by a neural network. Alongside lateral and longitudinal control, many other tasks carry great importance in robotic navigation, such as depth and ego-motion estimation, where different types of methods have been involved, reaching the potential of predicting accurate dense depth maps and vehicle trajectories only from stereo images. In this section, we offer a brief overview of the existing literature on end-to-end approaches for autonomous driving, environment perception, and ego-motion estimation.

### 2.1. Steering

A common approach for learning an end-to-end driving policy is via imitation learning from human expert demonstrations. One of the first successful deployments of a self-driving policy into the real-world environment is presented by [7], showing that a convolutional neural network (CNN) can map raw pixels from a single front-facing camera to steering commands. Human driving skills are replicated on local roads, highways, and areas with unclear visual guidance, such as parking lots and unpaved roads, achieving an autonomy of 98% and demonstrating a vast capability to generalize.

Rather than learning a single objective, some approaches include secondary tasks to enhance the main motion planning module [11,12]. Segmentation, depth, and motion estimation can provide useful information to improve autonomous driving tasks as they represent a smaller state space, thus allowing for better generalization. Depth estimation can improve road detections as it lies below the horizon line, and objects like buildings, trees, poles, etc., can be distinguished by their geometry. On the other hand, motion can capture highly dynamic traffic participants such as vehicles and pedestrians, while semantic segmentation can improve the overall understanding of the environment. Other recent approaches followed a similar idea and investigated the impact of different pre-trained perception modules on the model’s performance by either fusing RGB, depth, and semantic segmentation into a latent space [2], or providing a fully stacked input representation [13].

By combining sensor fusion and end-to-end learning, the authors of [14] replaced the classic raw RGB observations of the scene with a top-down synthetic view of the surroundings. Although the representation allows for flexibility in designing and optimizing auxiliary tasks, the method itself requires expensive perception hardware and relies completely on the quality of the representation.

In addition to the supervised learning paradigm, reinforcement learning has been successfully applied in autonomous driving. The authors of [15] designed a deep reinforcement learning algorithm to learn a driving policy within a handful of training episodes. Instead of physically interacting with the real-world environment, ref. [16] proposed a data-driven simulator and a training pipeline capable of learning vehicle control policy using sparse rewards, which successfully transferred to the real-world environment without additional fine-tuning.

One of the main challenges regarding the learning of steering commands, especially in end-to-end methods is the inability to correctly react to scenarios that are not previously met in the training data and the exponential amount of data needed to be added for every parameter of the environment.

### 2.2. Path Generation

The authors of [8] proposed a CNN cost model that learns to predict a cost map, given a single RGB input image. The cost map generation process relies on the IMU collected data which provides the relative pose of the vehicle between two consecutive frames and it is further used as an input for a trajectory optimization module that uses model predictive control.

Following a similar principle, the work of [9] described a weakly-supervised approach to generate labels for drivable path segmentation. The labeling process relies on several sensors such as IMU for the odometry data and a LIDAR for object detection by using data from the point cloud. The self-labeling pipeline projects future location points into the current frame, and an obstacle detection mechanism limits the length of the generated path. Finally, the training procedure follows a supervised paradigm where a CNN module learns to predict the drivable path from a single RGB input image.

Further work shows different improvements brought to the trajectory segmentation models, including the usage of Bayesian encoder-decoder networks [17] which can add explainability to the detection. Another improvement comes from the fusion between trajectory segmentation and steering angle estimation [18] or leveraging optical flow in a sequence-based approach for trajectory segmentation with training on labels obtained by projecting the location of the vehicle, obtained using GPS into each frame [19].

In off-road environments, terrain segmentation methods can provide valuable input and similar approaches that can be adapted for the path generation task. It has been shown how using image and laser data can be used to generate datasets for the unsupervised learning of traversable area segmentation [20]. Another method [21] depicts again how to generate data for learning terrain property in a self-supervised manner by using the force-torque signal of a robot, while using LIDAR, a bird eye’s view network [22] can be deployed to predict terrain classes. In supervised scenarios, where sensors like IMU, LIDAR and GPS are involved, path generation does not pose as many difficulties as in unsupervised or self-supervised approaches, mostly due to errors that accumulate at each step during ego-motion inference.

### 2.3. Learning Depth and Ego-Motion

Depth prediction from an RGB image has been a long-standing problem in computer vision. Although many approaches using stereo-vision [23,24] exist, recent work has been focused on using deep neural networks for depth prediction under supervised or self-supervised settings. Supervised learning methods achieve impressive results [25,26,27,28,29] but do not transfer over multiple datasets and depend on the available sensory depth data. Conventional approaches for unsupervised methods revolve around mapping source images to a target image and comparing them via a photometric consistency loss, involving view synthesis as the supervision signal. Such methods either require simultaneously training depth and pose predictors from one frame to another [5,30,31,32] or making use of calibrated cameras [33]. Other approaches also bring the optical flow learning problem alongside depth and ego-motion [34,35]. Complementary to the previous methods, ref. [36] proposed a change in the objective function, by comparing reconstructed point clouds at different time steps using the Iterative Closest Point (ICP) algorithm [37,38,39]. Another improvement that relies on tweaking the objective function for depth estimation networks is shown by the authors of [40], who present a quadtree-constrained photometric loss.

Ref. [41] introduced a novel approach to predict ego-motion and depth from videos captured by the random camera by simultaneously learning their intrinsic parameters, which were part of required a priori knowledge in the previous methods. The qualitative evaluation on a public video recorded by hand-held cameras proved the generalization power of the approach.

In addition to providing a correct depth estimation up to scale, previous methods suffer from scale-inconsistent results. Ref. [10] proposed to minimize the inconsistency between the projected and estimated depths for two consecutive frames, which leads to a scale consistent prediction across the entire video.

In a survey regarding single image depth estimation, ref. [42] present a broad range of state-of-the-art methods which also cover the category of self-supervised joint learning of depth end ego-motion.

## 3. Datasets

We selected two publicly available datasets to train and test our models, which comply with the needed requirements for evaluating our methods. First, we must be able to assess the reliability of the steering labeling, therefore a dataset with the ground-truth labels for steering has to be used. Secondly, the evaluation of the path labeling method requires a dataset with available ground truth data for this path or data that can be used to generate the path, such as the ego-motion labels. In this section, we describe the datasets we used for training and evaluating the proposed methods.

**UPB campus dataset:** the UPB campus dataset [43] was collected on the streets belonging to the campus of University Politehnica of Bucharest and presented in a work focused on defining a guideline on collecting, processing and annotating a self-driving dataset.

It contains 408 videos of normal driving, covering a distance of 72.7 km in 254 min, with 21% being recorded in low light conditions. Splitting the dataset into four discrete commands (go straight, stop, turn left and turn right) leads to a class distribution of 86.67% for straight driving, 5.45% for stop, and 4.45% for turning left, and 3.23% for turning right. For our experiments, we down-sampled the videos to 10 frames per second (FPS) and we split the dataset in two disjoint sets, 80% train, 20% test, and distributed in the two groups such that the geographical overlap is minimal, as shown in Figure 1. This split provides a fair evaluation and tests the model’s ability to generalize in a previously unseen environment.

**KITTI odometry dataset:** the KITTI odometry dataset [44] provides 11 driving sequences with ground truth trajectories and poses and another 11 sequences without any annotation. It contains driving sequences from various types of urban scenes. This dataset will be used for three main tasks. First of all, we will generate self-supervised path labels using the estimated ego-motion. Secondly, these path labels will be used as the target of a segmentation network that will learn to predict them when taking a single frame as an input. Lastly, we will benchmark the segmentation network by comparing its path predictions with the path obtained from the ground truth ego-motion of the dataset and with the path obtained from the predicted ego-motion.

The frames from this dataset are sampled at 10 FPS and we scaled them to 256 × 832, before cropping 96 pixels from each side to reach a final size of 256 × 640 for the path labeling using the estimated poses and the training of the segmentation neural network.

For this dataset, the split contains the sequences 3, 6, 9 and 10 for testing, 4, 11 and 19 for validation and the rest for training. We chose this dataset for training and evaluating our segmentation model against both the path labels generated using the ground truth ego-motion and the path labels obtained with the ego-motion network.

## 4. Proposed Method

In this section, we present the methods we employed for solving the steering labels generation, path labeling, path segmentation, followed by the implementation of the steering network and the improvements we proposed for enhancing the predictions of the network by leveraging the Ackermann steering geometry. We provided a detailed description of the mechanisms for each method, the models’ architecture, and the experimental setup we configured for both training and evaluation. The pipeline starts with the depth and ego-motion prediction network, which is trained on the UPB campus dataset and is followed by the scaling factor estimation, a step that is needed for recovering an accurate description of the motion. The obtained motion prediction is then utilized for two separate annotation tasks: the labeling of the steering and the labeling of the followed path, for each frame. At the end of the section, we also present a method of combining the path labeling into the steering prediction network for improved autonomy results.

A conceptual chart of the entire pipeline can be visualized in Figure 2.

### 4.1. Scaling Factor Estimation

One disadvantage of monocular depth estimation is that we can only recover the actual depth up to scale. To find the scaling factor automatically, we proposed a heuristic method that uses the position of the acquisition camera. We first transform the pixel coordinates to camera coordinates by the following sequence of operations:(1)$$\left[\begin{array}{c} p_{x}\\ p_{y} \end{array}\right]\overset{hom}{\to }\left[\begin{array}{c} p_{x}\\ p_{y}\\ 1 \end{array}\right]\overset{z}{\to }\left[\begin{array}{c} p_{x}\cdot z\\ p_{y}\cdot z\\ z \end{array}\right]\overset{M^{-1}}{\to }\left[\begin{array}{c} x\\ y\\ z \end{array}\right]$$
where $\left[p_{x},p_{y}\right]$ denote the pixel coordinates, [x,y,z] denote the world coordinates, $M_{3\times 3}$ is the intrinsic camera matrix.

We sampled the *y* coordinates in the pixel space from a rectangular box corresponding to the road patch in front of the vehicle. Let (h,w) be the height and the width of the image. In our experiments, the box is delimited by (h−10,h)×(w/2−50,w/2+50), from which we take the median value of the *y* coordinate ($m_{y}$). The extrinsic camera matrix provides the *y* coordinate of the camera denoted by $c_{y}$. We estimate the scaling factor as $f=\frac{c_{y}}{m_{y}}$.

### 4.2. Steering Labeling

The majority of deep learning models trained for the steering task and tested in the real-world environment use a single input observation [7,16] or short time dependencies, up to two frames [2]. Additional time dependencies have been reported to degrade the network’s performance [2] as the models started to capture spurious correlations [14], therefore, we opt for a method that uses single input observations and labels, since we consider it possesses a better ability to capture the actual reasoning behind an action instead of only correlating it to a set of past actions which can depict false reasons for taking a decision.

Leveraging the limited time horizon of the input, we proposed to generate steering labels by using the pose estimation network. The pose estimation model trained jointly with the depth prediction network provides the output as a six-dimensional tensor, corresponding to the six degrees of freedom of the middle-top camera, encoded as $\left[t_{x},t_{y},t_{z},r_{x},r_{y},r_{z}\right]$. The first three values represent the translations that happen on the three dimensional axes, while the last values are the rotations of the vehicle-mounted camera on these axes. We computed the turning radius as $R=s\cdot \sqrt{\frac{t_{x}^{2}+t_{y}^{2}}{2(1-\cos\left(r_{y}\right))}}$, where *s* is the scaling factor, and left/right turns are determined by the sign of $r_{y}$. This formula for the turning radius can be obtained using the law of cosines. Considering the distance between the positions of the vehicle at two different time steps is given as $d=\sqrt{t_{x}^{2}+t_{z}^{2}}$ and $r_{y}$ is the rotation angle around the *y* axis, then the circle arc on which the vehicle moves between the two different points will have an angle of $r_{y}$. Therefore, the distance *d* can also be expressed as $d=\sqrt{R^{2}+R^{2}-2\cdot R\cdot R\cdot \cos r_{y}}$. A visual explanation for how the vehicle moves on the circle arc from point $P_{0}$ and $P_{1}$ and the computation of the turning radius can be seen in Figure 3. The turning radius is further used to generate the targets for the learning process of the steering prediction task, as it is described in Section 4.4.

### 4.3. Path Labeling

Unlike the previous methods used for detecting and labeling the path, which are not completely self-supervised, we propose a self-supervised approach that combines previous proposed ideas for pose estimation [5,10,41] and path labeling to be further used in generating path labels for each input frame.

Ref. [9] showed how the relative pose between consecutive frames can be used to generate path labels. However, this approach involves the usage of an IMU sensor for recording the pose alongside the vehicle’s path. Therefore, in this section, we analyze the methods used for obtaining ego-motion (pose) estimates, given as a rotation matrix and a translation vector, from monocular images in a self-supervised manner, directly from RGB inputs, without the need for additional sensors. These methods are valuable for giving the possibility to be applied on videos in the wild and further generate usable data from them, such as the path labels in our case, without being limited to cases where vehicles are equipped with additional sensors. Therefore, this largely broadens the amount of driving data that can be used for different learning problems, such as the steering prediction, in the case of this work.

The self-supervised path labeling pipeline consists of two main steps. During the first step, two neural networks, described in Section 4.3.1, are trained to jointly learn depth and ego-motion from sequences of monocular frames. Once the networks are trained, the pose between different frames of a video can be predicted and we collect all the poses from each video, relatively to the time step of the first frame. The second step of the pipeline begins with iterating through each frame of the video and taking the next frames starting immediately after it, computing the relative poses between the camera position in the current frame and the same position from all the next selected frames, then projecting the contact points between the vehicle’s wheels and the ground. Once we have these points, we can construct a dataset of labels for the path and then train a segmentation model to learn such paths and to generate new ones when exposed to previously unseen data.

#### 4.3.1. Learning Ego-Motion

The first component of the pipeline is represented by the models that jointly learn depth and ego-motion in a self-supervised manner, from monocular frames. The approach is the one used in [10], which employs two separate neural networks, one for predicting depth maps and one for ego-motion. The depth neural network is a DispResNet, which takes the DispNet architecture as in [45] and replaces the convolutional layers of the encoder with residual ones. Therefore, the encoder is made of a ResNet50 [46] and the decoder consists of convolutional blocks with 3 × 3 kernel sizes and a number of channels which decrease from 256 to 16 by a factor of two, over five layers.

Having a source and target frame, we aim for warping the source into the target, using the equation that ties together the two frames and the pose between them. Minimizing the differences between the target image and the warped source image using the estimated pose between the two images, means getting closer to the true values of the rotation and translation that project the source into the target. The solution to warping from source to target is to apply an inverse warp from target to source. Therefore, for each point of the target, we want to find the location that the point will have in the source. To do this, the rotation and translation estimated as the pose between the two frames are applied to the target frame after it was converted to 3D coordinates. After the motion transformation of the target frame to the space of the source frame using the rotation matrix and the translation vector, the target frame is converted back to pixel coordinates, which will take continuous values in [−1,1]. Since the values are continuous, a differentiable bilinear sampling mechanism is applied to interpolate the coordinate value and compute the actual corresponding pixel from the source frame. After this, the depth and the RGB pixel values of the source frame can be placed on the grid, since all the coordinate values will be integers.

Equation (Equation 2) represents the backbone of jointly learning depth and ego-motion in a self-supervised manner.
(2)$$h\left(p_{s}\right)=KT_{t\to s}D\left(p_{t}\right)K^{-1}h\left(p_{t}\right)$$

This represents the view synthesis process when given a source and a target frame. The notations are as follows: $h(p_{s})$ and $h(p_{t})$ represent the homogeneous pixel coordinates of the two views *s* for the source and *t* for the target, *K* is the intrinsic matrix of the camera, $T_{t\to s}$ represents the motion transformation between the target and the source frames and $D(p_{t})$ is the predicted depth of the target view. The equation ties the depth and ego-motion, therefore creating a task where both are jointly optimized during the back propagation process.

#### 4.3.2. Trajectory Generation

Once all the poses from a sequence of frames have been collected, it is possible to compute the relative pose between each pair of frames. This needs to be computed, since the pose outputs are all relative to the first frame of the sequence, which is considered an identity matrix. Using the relative pose, any arbitrary point from a time step can be projected into a frame from another time step, therefore, we select two points which are fixed for every frame and represent the contact points between the vehicle and the road. To determine the position of these points, we can use the extrinsic properties of the camera. After that, considering a frame at the *t* time steps and an integer value *k*, we project the two points from all the t+1 to t+k time steps into the frame at time step *t*. We will consider *P* as the set of all points we want to project. $Pose_{t,t+k}$ represents the relative pose between the frames at *t* and t+k, while $P_{t,t+k}$ denotes the projection of all the points in the set *P* from frame t+k to frame *t* and it can be computed according to Equation (Equation 3).
(3)$$P_{t,t+k}=Pose_{t,t+k}P$$

We considered 60 future points which were projected into the current frame using the pose and the camera parameters.

**Generating hard labels.** The hard labels use the initial generated labels and consists of two separate classes. The first class represents the path followed and the second class consists of all the pixels that belong to the rest of the scene. Figure 4 shows few samples of the generated hard labels.

**Generating soft labels.** To obtain the soft labels cost map for the segmentation process, we started from the initial hard labels generated at the previous step as a two colors image and obtained a blurred version of them by applying a Gaussian filter with a kernel size of 31 × 31 and a standard deviation of 51. After applying the blurring filter, the obtained values are normalized between [0,1]. Figure 5 depicts the results of the soft labels generation process starting from a hard label given for a specific input.

The intuition behind using a soft version of the labels lies in the possibility to train a segmentation network with these labels as a target and then apply a threshold over the segmentation map that results as an output. This threshold would suggest how strong the network’s belief in the prediction should be, to consider this prediction as part of the predicted path.

### 4.4. Steering Training

We adapted the work of [7] to a single camera constraint. We enrich the training dataset with synthetic scenarios obtained by simulating displacements of the car using the 2D perspective augmentation of the input frames. Additionally, we provide the corrected steering command for the augmented frame, such that the simulated vehicle will take an action that will correct its position to match the position of the original one, after a given distance. Similar to [11], to capture multimodalities (e.g., intersection), our model outputs a discrete distribution over the future relative courses. This approach prevents from penalizing incorrectly the networks decision where multiple future trajectories are plausible.

We used imitation learning to predict the instantaneous road curvature recorded by the steering acquisition sensor or predicted by the pose-estimation module from two consecutive frames sampled at 0.1 s. The road curvature is defined as $\frac{1}{R}$, where *R* is the turning radius. As previously remarked in [7], this representation avoids singularity and furthermore is independent of the car geometry and covers the bounded interval $\left[-\frac{1}{R_{min}},\frac{1}{R_{min}}\right]$, where $R_{min}$ represents the minimum turning radius. In our current setup, we considered $r_{min}=5$ [m]. We modeled the target by a normal distribution centered in the ground truth road curvature and we fixed a standard deviation of 0.01. After investigating the histogram of the ground truth road curvatures, we decided to discretize the output into 401 bins. One decimal precision has proven to be necessary to capture multimodalities, indicating different routes that the car can take in a complex scenario (e.g., intersection).

The architecture of our model follows a ResNet18 [47] structure. The training procedure optimized the Kullback–Leibler (KL) divergence between the ground truth distribution and the predicted distribution. We trained our models for five epochs using the SGD optimizer with a batch-size of 256 and an initial learning rate of 0.1, decayed by a factor of 0.5 every epoch. We regularized the model by including an dropout of 0.5 before the output layer, a weight decay of 0.001.

To address the systematic bias from the dataset (86.6% of the time, the car goes in a straight direction) we experimented with data balancing by splitting the target into five classes, according to the absolute curvature. This split allows us to capture a broad enough spectrum of turning scenarios, covering both wide and tight turns and also straight line driving. More splits could allow better exposure of the models to extreme corner cases, but we considered this number to be enough to mitigate the bias that exists in the dataset. The weight of each class is inversely proportional to the number of examples that belong to it and during training, we sampled with replacements according to the distribution of weights. We choose five classes to capture representative samples for different types of driving scenarios according to the road curvature. Equation (Equation 4) emphasizes the formula used for weighting each sample in the dataset. Here, $w_{i}$ represents the weight of the sample *i*, *N* is the total number of samples in the dataset, #classes is the number of classes we selected for discretizing the steering values, C(i) represents the class of sample *i* and |·| is the operator that denotes the cardinality of a set.
(4)$$w_{i}=\frac{N}{\#classes\cdot |C(i)|}$$

Following the work of [7], our training pipeline augments 50% of the data instances with recoveries from erratic driving by applying perspective transformations that correspond to random translations and/or rotations sampled uniformly (both positive and negative values are considered) considering the following cases:if only a translation is applied, we sampled the shift uniformly from [0.5,1.5] [m]if only a rotation is applied, we sampled the rotation uniformly from [0.05,0.25] [rad].if a translation and a rotation is applied, we sampled uniformly from [0.5,1.2]×[0.05,0.12] [m, rad]

### 4.5. Path Segmentation

The approach we employ for path generation is based on semantic segmentation. We generated a dataset using the path labels and considered two separate variants for them: hard labels and soft labels.

As a segmentation model for the task of path generation, we chose DeepLabv3 [48] with a ResNet50 [47] backbone, which was pre-trained on ImageNet [49]. The loss functions we employ in the training process are cross entropy for the model trained on the hard labels and binary cross entropy with logits for the model trained with the targets defined by the soft labels cost maps. As an optimizer, we select SGD, with an initial learning rate of 0.0001, momentum of 0.9 and a weight decay of 0.0001. The model was trained for 50 epochs and the selected batch size was eight.

During training, a balancing technique was applied, since the datasets are biased and the predominant scenario emphasizes the vehicle moving forward. Therefore, for a generated path, we use the relative poses of the camera points between each pair of consecutive frames and then compute the rotation angle around the *Y* axis and sum all the obtained rotation angles along the followed path. Based on this sum, each frame was categorized as one of five different classes and during the training, each category was selected with the same probability. The categories are defined by angles between (−∞,−28), [−28,−8), [−8,−8](8,28] and (28,∞). These values have been chosen to represent different types of courses, varying from tight turns to left and right, to wider turns in both directions and straight line driving.

### 4.6. Improving Steering Predictions Using Path Labels

Furthermore, after having both the base steering generation networks and the path segmentation ones, we designed two additional methods to test if the performance of the steering generation can be improved.

**Encoded ROI**: the first idea suggests using the segmentation path labels as a ROI map which is fed as an additional input to the steering network. The steering network remains the same as the one described in Section 4.4, but the main difference is adding a convolutional encoder for the ROI map, whose output is then multiplied with the output of the first convolution in the ResNet, before being forwarded as input to the ResNet blocks.

Considering the initial steering network as a ResNet backbone and a fully connected layer for obtaining a course distribution output, we can express this process through Equation (Equation 5):(5)y=L(R(C(x)))
where *x* is the input frame, *C* denotes a convolutional encoder, *R* represents the ResNet backbone for feature extraction and *L* is the linear layer used for the output distribution generation.

The process of obtaining the output distribution for the encoded ROI solution is shown in Equation (Equation 6):(6)y=L(R(C(x)·C(S(x)))
where we used the same notations as in Equation (Equation 5) and *S* represents the segmentation network which is applied on the image input and (·) is a tensor multiplication operator.

**Label guidance:** the second idea employs the Ackermann steering model, which is used to generate a vehicle’s trajectory starting from a wheel angle input and a turning distance. Having the output distribution generated by the steering prediction network, we found the peaks of the distribution and then computed the corresponding wheel angle for each peak. Furthermore, for each of these wheel angles, we generated a path using the Ackermann geometry and then used the segmentation results to compute an overlapping score between the two. In the case of the soft segmentation method, this score is computed as the sum of the segmentation score for each pixel that lies in the geometrically generated path, while for the hard label segmentation, the score is represented by the mean intersection over union between the segmentation network output and the geometrically generated path. The path with the highest score was selected and its corresponding wheel angle was converted to steering angle and chosen as the command to be followed by the vehicle.

## 5. Evaluation

In this section, we evaluate our automatic labeling techniques, for the generation of both steering labels and path labels. We also analyze the influence of the deducted ego-motion scaling factor over the quality of the steering labels. Furthermore, we evaluate and compare the performance of a steering network when trained on the generated steering labels against the same network trained on the ground truth steering. The final part of this section is represented by the results obtained for the task of path segmentation.

### 5.1. Hardware Setup

For training all our neural networks, we have used the PyTorch deep learning framework [50] l alongside CUDA version 12.1 for GPU acceleration on a Linux machine with an Intel Core i7-6800K CPU, 32 GB of DDR4 RAM at 2400 MHz and loaded the models on a single NVIDIA GTX 1080Ti with 12 GB of memory.

### 5.2. Scaling Factor

By applying the heuristic described in Section 4.1 to our car setup, we obtained an average scaling factor of 32.8. To analyze the accuracy of our scaling factor estimation heuristic we proceeded as follows: initially, we have extracted the unscaled turning radius from the pose estimation module; then we swept over the interval [10,60] of potential scaling factors, and corrected our predictions. Finally we computed the mean squared error (MSE) of the road curvature against the ground-truth labels and we reported the minimizer. For the straight-forward computation, we obtained a scaling factor value of 34.65. As depicted in Figure 6, we can observe that the error decreases fast until it reaches the minimizing value, and afterwards slowly increases. The slow increasing rate is attributed to the systematic bias present in the dataset (most of the time the car goes in a straight line). By increasing the scaling factor, we implicitly increase the turning radius, which corresponds to a decrease in the road curvatures. Thus most of the road curvature values oscillate around 0, which corresponds to the dominant value in our dataset. To address this issue, we debiased the MSE by computing a weighted average using the sampling probabilities for data balancing. Our later procedure provides in a clear turning point of MSE plot, resulting in a faster increasing rate after the minimum values is surpassed (see Figure 6). Our predicted value for the scaling factor is clipped between the biased and unbiased minimizers, showing that our method provides an accurate estimate. The same figure shows the difference between the same trajectory with the minimizers’ scale values, on the right.

### 5.3. Steering Labeling

We performed open-loop and closed-loop evaluation of our models in a video-based simulator. For the open-loop evaluation, we used KL divergence as an error metric between the ground truth and the output distributions over the next possible road curvature. Our closed-loop evaluation relies on a video-based simulator similar to [7,16]. Each simulation step provides a single view corresponding to the middle front facing camera, by applying the corresponding view port transformations that reflects the previous decision of the steering policy. In this way, the current observation received by the network is determined by its previous decisions, and thus we simulate the car’s movement through the environment as if it was released in the real world. Our implementations describes the car’s trajectory using the Ackermann steering model, and at every step we monitor the displacement between the virtual and the real car to identify a human intervention and penalize the model. We followed the same evaluation metric proposed in [7], by reporting the autonomy and the number of interventions. We consider an intervention when the displacement (translation and rotation) between the virtual and the real car surpasses a given threshold, set to 1.5 [m] and 0.2 [rad], respectively. Each intervention is penalized by 6 s.

To obtain a universal reference value for the sensitivity of our closed-loop evaluation system, where 86.67% corresponds to driving in a straight line, we considered as a baseline a model that which always predicts a future course of $0^{\circ }$. The relatively low performance of the baseline, 45% autonomy, suggests that our simulator is able to capture fine movements of the simulated vehicle.

Initially, we trained a model on the ground-truth data recorded by the steering acquisition sensor. We applied multiple training procedure in terms of data sampling and data augmentation, including: raw dataset, data balancing, perspective augmentation (shifts—corresponding to recoveries from erratic driving), and finally combining data balancing and shifts. During our training procedure, we monitored the open-loop performance on the validation dataset, but we did not find it to be informative and correlated with the closed-loop evaluation. Thus, we reported the evaluation results for the model that achieves the best closed-loop performance on the validation dataset. The results of our experiments are summarized in Table 1. For our current setup, data balancing slightly degraded the performance of our model. We attribute that loss to the systematic bias that also exists in the validation dataset, where a balanced model could predict more turning commands, therefore decreasing the overall autonomy on a dataset where the bias emphasizes going in a straight line. By weighting the five class equally, our model can forget about the actual trajectory distribution. UPB dataset [43] lacks of diversity in terms of intersection and curved roads, and we suspect that our model has difficulties in generalizing. We leave the analysis of other balancing strategies for future work. Our best model achieves a 79% autonomy, equivalent to 134 interventions over approximately 51 min of recorded driving (without accounting the time penalty for each intervention). The 2D perspective augmentation introduced in the training dataset boosted the autonomy of the model with 9% and reduced the number of interventions with 84. Most of the time, the model struggles in intersections since we did not condition the policy to follow a predefined trajectory. From the manual inspection of the simulations, we observed that the model learned to avoid obstacles such as stationary cars, fences in construction areas and pedestrians that walk on the drivable area. Moreover, since our model does not account for past dependencies, the model has difficulties returning to the correct lane when overtaking an obstacles, requiring human intervention.

We applied the same training procedure by replacing the ground-truth road curvature with the one resulted from the pose estimation module to evaluate the feasibility of our proposed method. In Table 1, we summarized the open-loop evaluation against the ground-truth and pose estimation labels and the and the closed-loop evaluations against the ground-truth labels. Similar to our previous experiment, we reported the best closed-loop performance on the validation dataset. Our best model achieves a 77% autonomy, equivalent to 159 intervention. The 2% decrease of the model’s performance from its corresponding ground-truth counterpart demonstrates that the pose-estimation module provides accurate labels, achieving competitive results against the collected sensory data.

Figure 7 illustrates previously described behavior of our models and the location of the intervention points.

### 5.4. Scale Sensitivity

One limitation of our pose estimation labeling pipeline is the recovery of the correct scaling factor. We analyzed the impact of the estimated scaling value on the steering network’s performance by training multiple models on different pose estimation labels obtained by varying the scaling factor between 22.8 and 42.8. We used the same training pipeline as described in Section 4.4, and we reported the open-loop and closed-loop evaluation of the models that achieved the best autonomy on the validation dataset. Our results are summarized in Table 2.

In terms of the open-loop evaluation against the ground-truth labels, as suggested by the downward-upward trend captured by our results, a better estimate of the scaling factor results in a lower error metric. On the other hand, the open-loop evaluation against pose estimation labels on the validation dataset does not offer any relevant information. As previously emphasized in Section 4.1, increasing the scaling leads to an increase in the turning radius, which implicitly results in a decrease in road curvatures. Thus, an overestimation of the scaling factor induces a bias in the constructed labels (almost a straight trajectory), which could be easily learned by a model, as suggested by the downward trend captured in our results.

Similarly to the open-loop evaluation against the ground-truth labels, we observe a downward-upward trend in the closed-loop performance. The average autonomy over the scaling factor sweep is 76.11% with a standard deviation of 1.19%, suggesting that the model learns a robust policy even for inaccurate scaling factors. Moreover, the model achieved the best autonomy of 77% for a range of scale values, between 30.3 and 37.8, an interval which contains our heuristically estimated value of 32.8. This again shows that our method for selecting the scaling factor is appropriate, since it obtains the highest autonomy with a small number of interventions higher than models trained with labels obtained from other scaling factors.

### 5.5. Segmentation

After the path labels have been obtained, we trained the segmentation model on both the KITTI odometry and UPB datasets. We define five different model alternatives: one which is trained with the hard labels as targets and four which are trained with the soft labels and we will denote them as below:**HL:** the model trained with the targets defined by the hard labels.**SL Logits:** the model trained with the targets defined by the soft labels, where a threshold is applied directly of the output of the model.**SL Sigmoid:** as SL Logits, but a Sigmoid operation is applied on the output logits.**SL Softmax:** as SL Logits, but a Softmax operation is applied on the output logits.**SL Tanh:** as SL Logits, but a Tanh operation is applied on the output logits.

To find the best threshold for the soft labels evaluation, we selected an interval and performed a search for each one of the four variants of the model. This search employs running the model on the testing split of the UPB campus dataset and then selecting the threshold that leads to the highest mIOU. The setup for this search follows the cases:**SL Logits:** search interval [−1,0.5], step size 0.025, with the best threshold at 0.15.**SL Sigmoid:** search interval [0.1,1], step size 0.025, with the best threshold at 0.425.**SL Softmax:** search interval [0.0025,0.035], step size 0.0025, with the best threshold at 0.0075.**SL Tanh:** search interval [−0.25,0.25], step size 0.025, with the best threshold at −0.15.

For evaluating the path segmentation, we chose two popular metrics: accuracy (Acc) and mean intersection over union (mIOU). Accuracy represents the percentage of pixels that were correctly classified, and the formula for this metric can be seen in Equation (Equation 7). In this equation, the following notations have been used: TP for true positives, TN for true negatives, FP for false positives and FN for false negatives. In our situation, true positive are pixels that are correctly identified to belong to the path class, true negatives are pixels that are correctly identified to belong to the rest of the scene, false positives are pixels that are incorrectly identified as belonging in the path class and false negatives are represented by the pixels incorrectly identified as belonging to the rest of the scene.
(7)$$accuracy=\frac{TP+TN}{TP+TN+FP+FN}$$

The second metric, *mIOU* can be described as the ratio between the intersection of the target with the prediction and the union of the two, averaged over all classes. Since we only have one class, denoting the followed path, the intersection represents the pixels where the ground-truth label of the path and the predicted label are overlapped, while the union represents the pixels where either the ground truth or the prediction is present. Equation (Equation 8) describes the *mIOU* metric for segmentation.
(8)$$mIOU=\frac{predicted\cap target}{predicted\cup target}$$

Table 3 shows the results of the segmentation process on different setups, where the threshold values for the soft label models have been selected as described above. The first column, denoted as KITTI SSL shows the performance on the KITTI dataset, after we compared the segmentation results of different variations of the model with the labels we obtained in a self-supervised manner. The second column, named KITTI GT, shows the segmentation results compared to the labels we obtained when using the ground truth poses from the dataset, while the final column shows the results of the segmentation on the UPB campus dataset against the self-supervised labels generated for this dataset. In the case of this dataset we could not evaluate against the labels obtained from the exact path of the vehicle, since there are no poses recorded during the data collection process.

Figure 8 shows several result samples of the HL model applied on the UPB campus dataset, while Figure 9 depicts the four variations of the SL model on a sample taken from the same dataset, revealing both the heatmap of the model output and the final output after applying a threshold over each heatmap.

We can see the models are able to generalize well for new driving sections that are not found in the training split of the dataset.

One feature we desired for the segmentation model was the ability to generate multiple paths in cases of reaching an intersection. There are some cases where the segmentation network generates multiple path possibilities, as seen in Figure 10.

The segmentation results show reliable path generation even for the cases of intersection, where the network is able to generate multiple paths, despite their absence from the ground truth ego-motion trajectories of the vehicle. However, there are also situations where the paths generated by the network show a lack of confidence in the trajectory followed by the vehicle, especially in turns where the road is not very well delimited.

### 5.6. Steering Improved

For the steering improvement evaluation, we use the steering network models trained on the dataset where the ground truth is generated by the steering labeling algorithm as described in Section 4.2 and we want to see if these improvements can guide these models to yield similar or better performance compared to the steering models trained on the dataset with the ground truth steering labels as collected by the sensors.

Therefore, in this section, we introduce two new models: POSE+S+ROI and POSE+S+ACK. The first model corresponds to the first one described in Section 4.6. The ROI map we used consists of the soft segmentation ground-truth labels, while at inference time, the SL Sigmoid model was applied on the RGB frame and the results are fed into the steering network, alongside with the single RGB input. The second model starts from the POSE+S model, mentioned in Table 1 and uses the segmentation guidance as described in Section 4.6. Figure 11 shows an example of this process. On the left, the output distribution of the steering network is shown, the image in the center suggests the possible paths formed from the distribution peaks, while the image on the right shows the heatmap representation of the results after applying the path segmentation network.

We compared these two additional models with the GT+S and POSE+S models, the best performing models from the previous experiments. The results can be seen in Table 4. The model that uses the segmentation labels as ROI input performs worse than the same model without the additional input, however, when the base model is guided by the overlap score between the Ackermann paths and the segmentaion labels, it even outperforms the model that is trained on the ground truth steering labels.

## 6. Conclusions

We have presented a fully self-supervised labeling pipeline for an autonomous driving dataset in the context of steering control and path labeling. We applied our method on two open-source driving datasets [43,44], and we proved the robustness and accuracy by achieving competitive results against the ground-truth labeling counterpart. Consequently, this shows that leveraging self-labeled data for learning a steering model can be almost as reliable as using the ground truth data. Additionally, our heuristic for determining a scaling factor for the depth and ego-motion resulted from the self-supervised learning pipeline has proven to be accurate and simple, which shows promising results in adapting the ego-motion and depth estimation to a real-world scale.

We consider that our current proposal can address two of the main limitations of the end-to-end approach regarding the amount and the diversity of available training data. Our method is flexible and applicable to a minimalistic data acquisition setup consisting of a single camera, which makes it suitable for data generation in the case of videos in the wild, significantly broadening the data that can be used for learning driving policies without being limited by the labels from a dataset or additional equipment which might not be available to everyone. Therefore, one potential line of research that we intend to investigate is to leverage the vast amount of online driving video resources to learn a robust driving policy.

In addition to providing two different self-supervised labeling techniques, one for steering labels and one for path masks, we have shown how the path segmentation network and steering geometry can be leveraged with the purpose of improving the decisions taken by the steering network, demonstrated by the fact that this setup surpassed the autonomy of the steering network when trained on ground truth steering data.

In summary, our contributions are as follows:a comprehensive performance evaluation of the unsupervised steering labeling procedure;a sensitivity analysis of the steering models performance relative to the estimated scaling factor;a pipeline that generates path labels for any given dataset, in a self-supervised manner;a performance evaluation of different variations of segmentation models trained on the generated paths;two different approaches used to improve the performance of the base models using the paths obtained through segmentation.

As for future work, we propose on pursuing a more robust way of generating the path labels, keeping the self-supervised approach, but also considering the obstacles that interfere in the trajectory followed by the vehicle. One approach involves the idea of discarding the path regions from the current frame that are not supposed to be visible due to occlusions and to achieve this, the disparity maps of all the frames alongside the trajectory can be used. Another possible research direction we want to follow through consists of eliminating the steering network from the decision making process and only provide steering outputs by leveraging the trajectory segmentation results and the Ackermann geometry. We are also commited to trying newer segmentation approaches, using state-of-the-art methods involving vision transformers and we plan on introducing a user-guided text embedding which represents the intention about the directions of the trajectories outputed by the segmentation model in intersection scenarios.

We believe that leveraging the vast amount of driving video recordings available on the internet will be a natural next step towards improving the capability of an autonomous driving system.

## Figures and Tables

**Figure 1 sensors-23-08473-f001:**
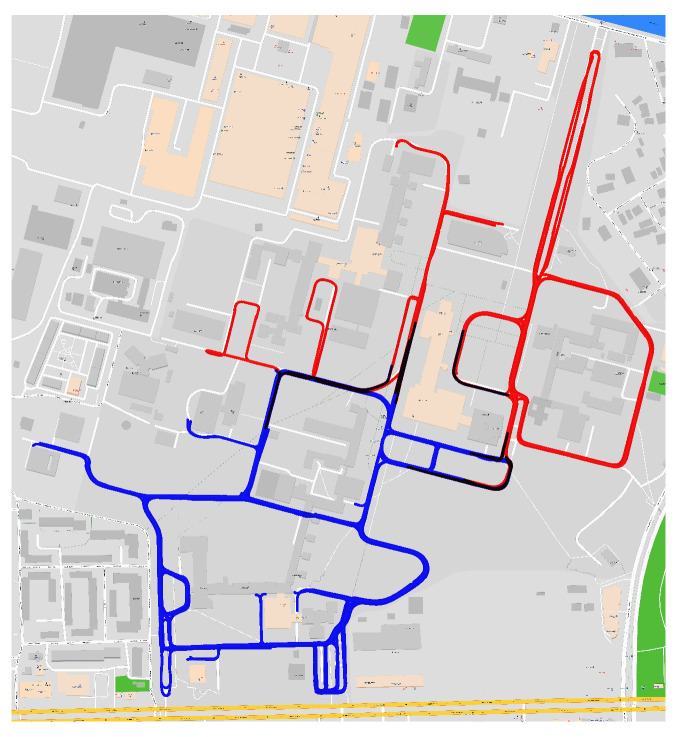
Train (blue), validation (red) dataset split. The overlap between the splits is depicted in black.

**Figure 2 sensors-23-08473-f002:**
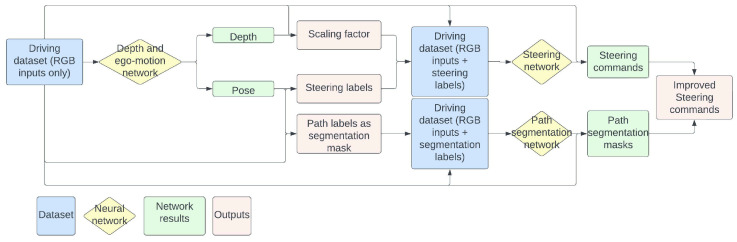
Pipeline to generate path labels in a self-supervised manner. Start with the RGB driving dataset, pass it through a depth end ego-motion network for obtaining a scaling factor, steering labels and path labels and then generate new datasets, one for a steering network and the other for path segmentation Finally, the segmentation and steering results are combined in a steering improvement module.

**Figure 3 sensors-23-08473-f003:**
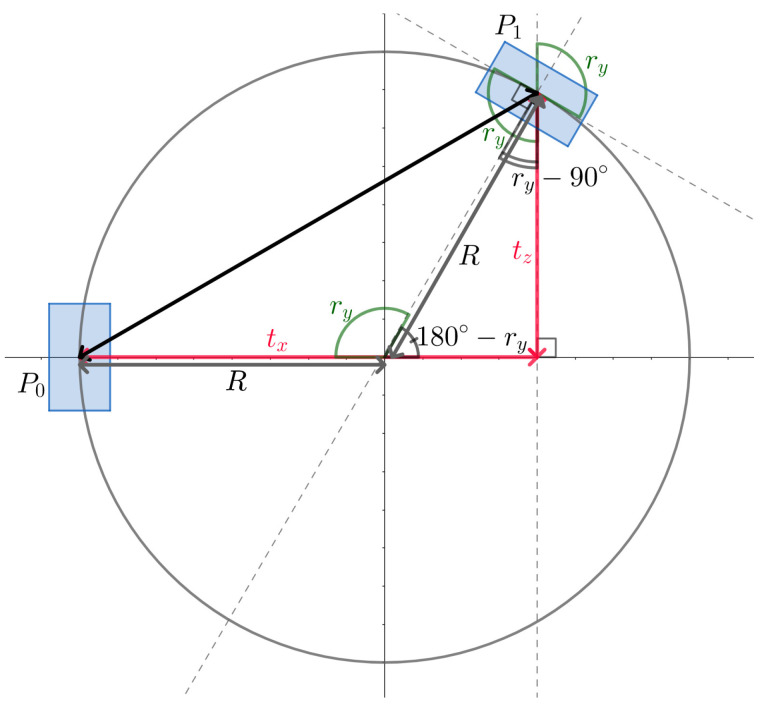
Computing the turning radius. Considering the initial and final positions of the vehicle, the turning radius can be computed using the law of cosines when the distance between the two different positions and the angle between the orientation of the vehicle in the two given positions are known.

**Figure 4 sensors-23-08473-f004:**
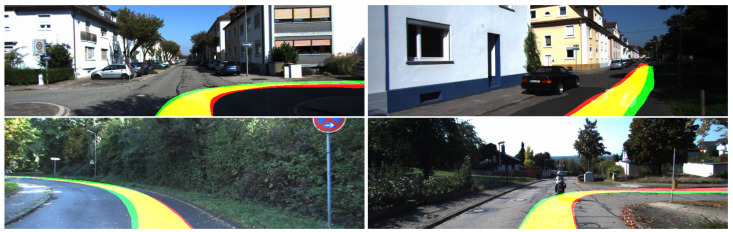
Generated samples for the path labeling process. Green depicts the path obtained using the estimated pose, red is the path obtained using the ground truth pose and yellow is the intersection between them.

**Figure 5 sensors-23-08473-f005:**
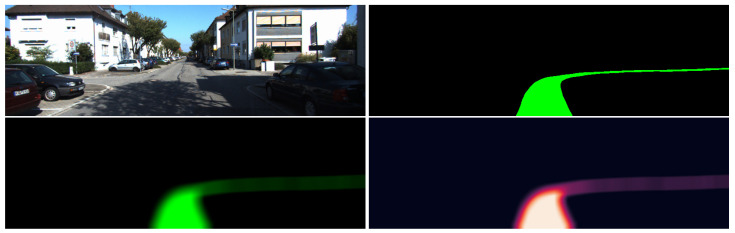
Soft label generation. After the hard label has been generated for the input frame, a Gaussian blur is applied over it. In the bottom right figure, the soft label can be seen as a heatmap, where the brightest pixels represent the highest values.

**Figure 6 sensors-23-08473-f006:**
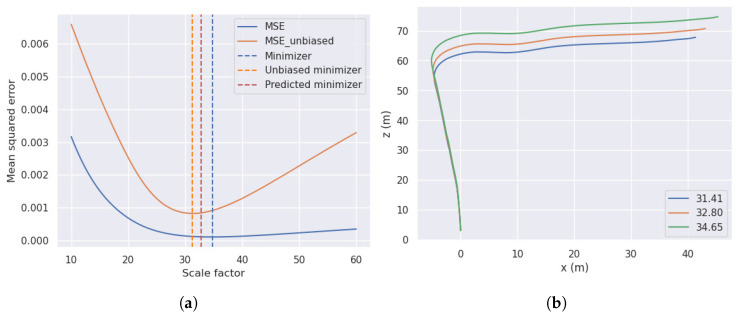
Influence of the scale factor over the MSE between the road curvature and the ground-truth labels (**a**) and over the aspect of the trajectory (**b**). (**a**) MSE of the road curvature against the ground-truth label for different scale factors. (**b**) Different trajectories constructed from the estimated pose and the corresponding scale factors.

**Figure 7 sensors-23-08473-f007:**
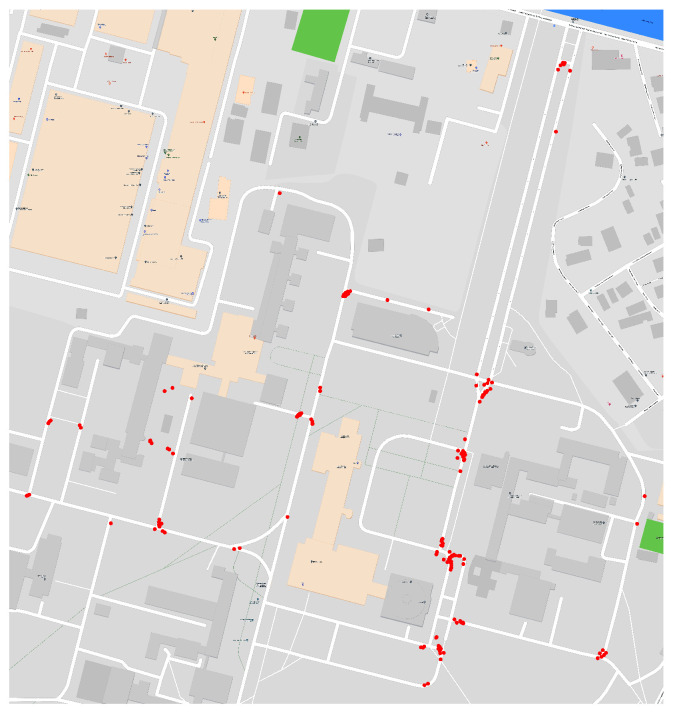
Intervention points for model trained with perspective augmentations.

**Figure 8 sensors-23-08473-f008:**
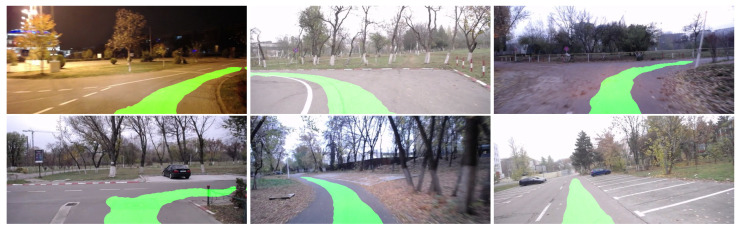
Segmentation samples for the UPB campus dataset obtained using the HL model.

**Figure 9 sensors-23-08473-f009:**
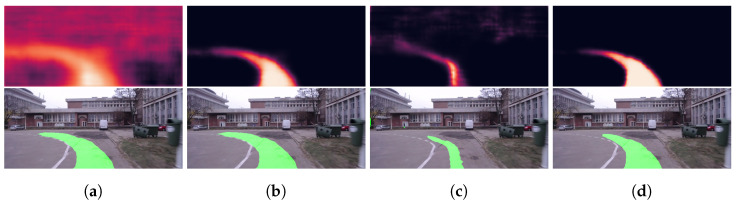
Segmentation samples for the UPB campus dataset and heatmap visualizations obtained using the SL models. (**a**) SL Logits. (**b**) SL Sigmoid. (**c**) SL Softmax. (**d**) SL Tanh.

**Figure 10 sensors-23-08473-f010:**
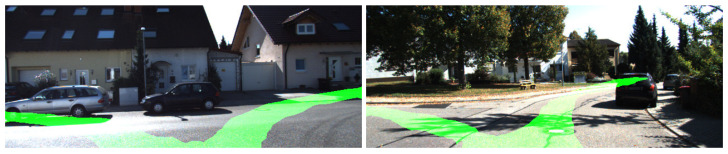
Multiple paths generated by the segmentation network.

**Figure 11 sensors-23-08473-f011:**
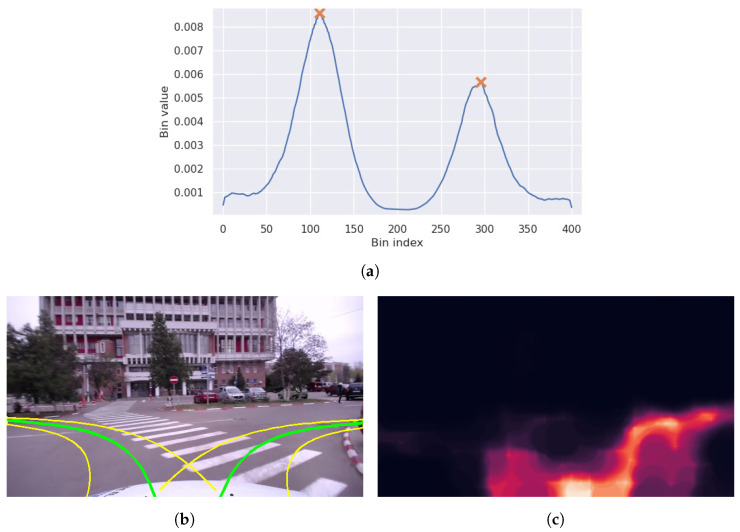
Guiding the steering network using the segmentation results. In this example, the steering network would choose the higher peak, from the left, but the segmentation model predicts a higher score for going to the right. (**a**) Steering network distribution output with the two detected peaks. (**b**) Paths generated with the Ackermann model from the distribution peaks. (**c**) Output of the path segmentation network.

**Table 1 sensors-23-08473-t001:** Evaluation results of the two experiments over 3060.16 [s]. Baseline corresponds to the model that predicts always $0^{\circ }$. Abbreviations: ground truth (GT), pose estimation (POSE), data balancing (B), perspective augmentation shifts (S), autonomy (A), number of interventions (NoI), not applicable (-), total time including penalties (TT).

	Open-Loop (KL)		Closed-Loop		
**Model**	**GT**	**POSE**	**A**	**NoI**	**TT [s]**
Baseline	4.05	-	0.45	623	6846.67
GT	0.49	-	0.70	221	4434.67
GT + B	0.56	-	0.67	254	4632.67
GT + S	0.51	-	**0.79**	**134**	**3912.67**
GT + BS	0.61	-	0.77	153	4026.67
POSE	0.54	0.58	0.65	284	4812.67
POSE + B	0.60	0.62	0.65	276	4764.67
POSE + S	0.55	0.62	**0.77**	**159**	**4062.67**
POSE + BS	0.64	0.70	0.75	172	4140.67

**Table 2 sensors-23-08473-t002:** Scale sensitivity. The results for our scale prediction are written in bold. Abbreviations: ground truth (GT), pose estimation (POSE), Kullback–Leibler divergence (KL) autonomy (A), number of interventions (NoI).

Scale	22.80	25.30	27.80	30.30	**32.80**	35.30	37.80	40.30	42.80
A	0.73	0.76	0.76	0.77	**0.77**	0.77	0.77	0.76	0.76
NoI	194	164	160	157	**159**	153	159	161	164
KL(GT)	0.65	0.61	0.59	0.59	**0.55**	0.54	0.57	0.58	0.60
KL(POSE)	0.78	0.76	0.70	0.69	**0.62**	0.58	0.58	0.54	0.52

**Table 3 sensors-23-08473-t003:** Evaluation results for the path segmentation given the two metrics: accuracy (Acc) and mean intersection over union (mIOU).

	KITTI SSL		KITTI GT		UPB SSL	
**Model**	**Acc (%)**	**mIOU (%)**	**Acc (%)**	**mIOU (%)**	**Acc (%)**	**mIOU (%)**
HL	98.20	**84.79**	98.22	**85.23**	97.40	**83.89**
SL Logits	98.15	66.18	**98.25**	67.01	**98.14**	70.32
SL Sigmoid	**98.24**	66.21	97.44	66.96	97.87	70.51
SL Softmax	96.98	42.36	95.85	42.06	95.13	36.37
SL Tanh	98.13	66.16	98.19	66.37	97.92	70.70

**Table 4 sensors-23-08473-t004:** Evaluation results of the two additional experiments. Abbreviations: ground truth (GT), pose estimation (POSE), perspective augmentation shifts (S), autonomy (A), number of interventions (NoI), total time including penalties (TT), encoded region of interest model (ROI) and Ackermann geometry guided model (ACK).

	Closed-Loop		
**Model**	**A**	**NoI**	**TT [s]**
GT + S	0.79	134	3912.67
POSE + S	0.77	159	4062.67
POSE + S + ROI	0.75	175	4158.67
POSE + S + ACK	**0.80**	**128**	**3876.67**

## Data Availability

Data available on request.

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
