# Peer review of "Self-Supervised Steering and Path Labeling for Autonomous Driving"

_sensors, 2023, doi:10.3390/s23208473_

Round 1

Reviewer 1 Report

The paper presents self-supervised steering and path labeling for autonomous driving. The development of the autonomous system demonstrates the authors' commitment to conducting thorough experiments. Several results have been obtained, but some revisions are needed to improve the paper quality.

1.       The investigation has been done very well. But it seems that the contribution of the paper is combination of two existing works on path labeling and steering. In my opinion it cannot be novelty of the paper. So, please provide the main contributions of the paper.

2.       Please highlight the key contributions of the study in the introduction part.

3.       Describe in detail the hardware system that you trained DL models.

Author Response

Hello, Thank you for the time to review our submission and for your comments. Please find bellow our answers. 1. The investigation has been done very well. But it seems that the contribution of the paper is combination of two existing works on path labeling and steering. In my opinion it cannot be novelty of the paper. So, please provide the main contributions of the paper. - Added the contributions to the end of the introduction section 2. Please highlight the key contributions of the study in the introduction part. - Added contributions to the end of the introduction section 3. Describe in detail the hardware system that you trained DL models. – Added section 5.1 which describes the hardware system

Reviewer 2 Report

1. The entire article has clear logic and fluent language, with a high overall completion level. It is recommended to accept.

2. Some illustrations can be further beautified.

3. Formula calculation is further detailed.

4. The format of Figures 9 and 11 needs to be modified.

5. The summary and future outlook section can be elaborated in more detail.

6. The purpose of this article is relatively novel and has certain research significance.

7. The references in this paper try to quote the literature in the past three or five years to show the progressiveness of this research.

The overall English language of the article is moderate, with few grammatical errors that require moderate editing.

Author Response

Hello, Thank you for the time to review our submission and for your comments. Please find bellow our answers. 2. Some illustrations can be further beautified. - We have modified figures 9 and 11 to make them nicer 3. Formula calculation is further detailed. - No action required as far as we understood 4. The format of Figures 9 and 11 needs to be modified. - Modified them to fit the text width 5. The summary and future outlook section can be elaborated in more detail. - Added more info in the conclusion; reformulated a phrase to be more explicit and added more to the future work

Reviewer 3 Report

In this work, the authors present an approach that solely uses monocular camera inputs to generate valuable data without any supervision. The main contributions involve a mechanism that can provide steering data annotations starting from unlabeled data alongside a different pipeline that generates path labels in a completely self-supervised manner. However, the following problem need to be improved.

(1) The English of paper is poor, please polish it again.

(2) Each variable in the equation should be explained clearly.

(3) More recently methods should be compared to show the advantages of the proposed method.

can be improved

Author Response

Hello, Thank you for the time to review our submission and for your comments. Please find bellow our answers. In this work, the authors present an approach that solely uses monocular camera inputs to generate valuable data without any supervision. The main contributions involve a mechanism that can provide steering data annotations starting from unlabeled data alongside a different pipeline that generates path labels in a completely self-supervised manner. However, the following problem need to be improved. (1) The English of paper is poor, please polish it again. We have reformulated some ambiguous phrases and have an extra check for English (2) Each variable in the equation should be explained clearly. We have checked all the explanations of variables for all the equations (3) More recently methods should be compared to show the advantages of the proposed method. To our knowledge, there is a lack of contributions on self-supervised steering label generation; Even though there is related work approaching the same task as we do, they don't detail their evaluation process enough for us to be able to compare the results on the same data splits. One other impediment when it comes to comparing our results with other work comes from the usage of our own dataset, which is not yet mentioned in work related to our topic

Round 2

Reviewer 3 Report

The respond is OK.